# Inhibition of Nitric Oxide Production, Oxidative Stress Prevention, and Probiotic Activity of Lactic Acid Bacteria Isolated from the Human Vagina and Fermented Food

**DOI:** 10.3390/microorganisms7040109

**Published:** 2019-04-23

**Authors:** Chang-Ho Kang, Seul Hwa Han, Jin-Seong Kim, YongGyeong Kim, Yulah Jeong, Hye Min Park, Nam-Soo Paek

**Affiliations:** MEDIOGEN, Co., Ltd., Seoul 04146, Korea; sulhwa0275@gmail.com (S.H.H.); rlawls195@naver.com (J.-S.K.); yongkyung@naver.com (Y.K.); jinhwa110931@gmail.com (Y.J.); hyeminpark95@naver.com (H.M.P.); nspaek@mediogen.co.kr (N.-S.P.)

**Keywords:** nitric oxide, lactic acid bacteria, DPPH, probiotics

## Abstract

In this study, lactic acid bacteria (LAB) with antioxidative and probiotic activities were isolated from the vaginas of Korean women and from fermented food. Among 34 isolated LAB strains, four strains (MG4221, MG4231, MG4261, and MG4270) exhibited inhibitory activity against nitric oxide production. The MG4221 and MG4270 strains were identified as *Lactobacillus plantarum*, and MG4231 and MG4261 were identified as *Lactobacillus fermentum*. These strains were able to tolerate pepsin and pancreatin, which is required for probiotic potential. The antioxidant effects of culture filtrates obtained from selected strains included 2,2-diphenyl-1-picryl-hydrazyl (DPPH) radical scavenging capacity. Most of the culture filtrates had effective DPPH scavenging activity.In conclusion, the selected strains have significant activities and are potentially applicable to the development of functional foods. These strains might also contribute to the prevention and control of several diseases associated with oxidative stress, when used as functional probiotics.

## 1. Introduction

Probiotics are defined as “living microorganisms that have health benefits beyond inherent basic nutrition” when consumed in certain quantities [1,2]. The prebiotic definition according to World Health Organization (WHO)states that “live microorganisms which, when administered in adequate amounts, confer a health benefit on the host” [3,4]. In 2011, a guidance document on scientific requirements for health claims has been published by European Food Safety Authority (EFSA) related to gut and immune function. This guide addresses the beneficial effects and outcome measures that are acceptable for substantiation of claims in gut health, probiotics, and prebiotics [5,6]. The beneficial effects of probiotics include the prevention and treatment of diarrhea, prevention of systemic infections, inflammatory bowel disease management, immunodeficiency, prevention and treatment of allergies, anticancer effects, and treatment of cholesterolemia [7,8,9]. Several therapeutic benefits have been addressed to probiotics-based functional foods including anticancer, hypoglycemic, antioxidant, immunomodulatory effects, diabetes, allergic diseases, and obesity [10,11,12,13,14]. Therefore, there is substantial medical and industrial interest in the isolation of new probiotic strains with health-promoting benefits [11].

Lactobacilli play an important role by producing compounds that inhibit the growth of potential pathogens, such as lactic acid, bacteriocins, and hydrogen peroxide [15,16,17]. The total bacteria isolated from healthy women showed at least 70% vaginal lactobacilli [18]. In the human vagina, the lactobacillus species have received considerable attention because of their protective and probiotic properties [19]. Moreover, the traditional fermented foods have been consumed by millions of people for thousands of years because of their beneficial properties [20]. Most of these fermented foods are enriched with bacteria, which has been well reported in previous studies [21,22,23].

Inflammation is a complex reaction of the vascular tissues to harmful stimuli, such as pathogens, damaged cells, or irritants. It is mediated by various signal molecules produced by macrophages, monocytes, and mast cells. Persistent inflammatory stimuli can lead to chronic inflammation [24]; therefore, the monocytes population is maintained and existing macrophages are bound there. Macrophages are particularly important for innate immunity, as they are involved in initial cellular response to microbial infection. Nitric oxide (NO), a radical produced from l-arginine via the action of NO synthase (NOS) [25], is one of the most important inflammatory mediators with roles in various biological processes including neurotransmission, immune defense, and apoptosis. Three isoforms of NOS have been identified, named according to their Ca^2+^calmodulin dependence [26] or tissue type.

The aim of this study was to isolate lactic acid bacteria with high probiotic potentiality in exerting antioxidant activity for counteracting oxidative stress. We isolated various lactic acid bacteria (LAB) and, after confirming their safety and identity, evaluated their functional characteristics with a focus on their potential application as beneficial probiotic strains

## 2. Materials and Methods

### 2.1. Isolation of Microorganisms

In this study, 34 LAB were isolated from the vaginas of healthy Korean women [27] and Kimchi, traditional Korean fermented food. The kimchi samples were diluted with sterilized phosphate-buffered saline (PBS) by serial dilution. A 0.1 mL aliquot of each dilution was plated onto de Man Rogosa Sharpe (MRS) agar (Difco, Detroit, MI, USA) to screen LAB strains. The plates were anaerobically incubated at 37 °C for 24 h. After incubation, the catalase-negative, gram-positive isolates were preliminarily identified as LAB and subsequently grown in MRS broth or agar. To indirectly confirm whether the isolates produced lactic acid, they were plated on MRS agar containing 0.1% bromocresol purple. Only yellow colonies were selected and the isolates were stored in 25% glycerol at −70 °C.

### 2.2. Cell Culture

Macrophage RAW 264.7 cells were purchased from the Korean Cell Line Bank (KCLB). These cells were grown at 37 °C in 5% CO_2_ in fully humidified air and sub-cultured every 3 days to a confluence of 95%. For routine sub-cultivation, Dulbecco’s modified eagle’s medium (DMEM, Gibco, Grand Island, NY, USA) supplemented with 10% fetal bovine serum (FBS, Gibco), penicillin (100 unit/mL), and streptomycin (100 μg/mL) was used.

### 2.3. Nitric Oxide Assays

NO formation was detected as an accumulation of nitrite, an indicator of NO synthesis in the culture medium by the Griess reaction [28]. RAW 264.7 cells were plated at 2 × 10⁵cells/well in 96-well plates and stimulated with 1 μg/mL Lipopolysaccharide (LPS), followed by the addition of isolated strains (10^7^ cells/well). After 24 h of incubation, the NO concentration was determined by measuring the amount of nitrite in the cell culture supernatant, using the Griess reagent. Absorbance at 550 nm was measured using a microplate reader. Fresh culture medium was used as the blank control in all experiments.

### 2.4. Biochemical Tests and Identification

Selected isolates were identified by Gram staining, conventional biochemical tests [29], and sequencing of the 16S ribosomal RNA gene using universal primers (518Fand 800R). PCR and sequencing were performed by Macrogen Co. (Daejeon, Korea). Sequence similarity between strains was analyzed by generating nucleotide alignments using Macrogen Alignment (https://dna.macrogen.com/kor/), a web-based tool for identification based on 16S rRNA gene sequences. Basic local alignment search tool (BLAST; https://blast.ncbi.nlm.nih.gov/Blast.cgi) was used to compare the sequences obtained in this study with available DNA sequences registered in the National Center for Biotechnology Information. A phylogenetic tree was constructed with the neighbor-joining method using MEGA 5.0 software(https://www.megasoftware.net/) [30].

### 2.5. Morphological Analysis of Selected Strains by SEM

Strain morphologies were analyzed using scanning electron microscope (SEM) (S-4200; Hitachi, Tokyo, Japan). After incubation at 37 °C for 24 h, the cells were washed twice and resuspended in 0.1M PBS (pH 7.2). Then, 5 ml of the suspension was transferred to cover glass and dried at room temperature for 3 min. Next, the cells were fixed with 2.5% (*w/v*) glutaraldehyde in PBS for 3 h and then incubated in PBS for 12 h. The cells were rinsed twice with PBS. Finally, the samples were dehydrated in ascending concentrations of ethanol (10%, 25%, 50%, 75%, 95%, and 100% *v/v*) and dried at room temperature.

### 2.6. Measurement of 2,2-Diphenyl-1-Picryl-Hydrazyl (DPPH) Scavenging Activity

DPPH free radical scavenging activity was measured according to the method described by Lee et al. [31], with slight modifications. A 400 μM DPPH solution in methanol was prepared. Then, 2 mL of this solution was added to 2 mL of each resuspended strain. After incubation at room temperature for 30 min, the absorbance was measured at 517 nm. Two milliliters of methanol mixed with 2 mL of DPPH solution was used as the control. The inhibition of DPPH (expressed as a percentage) was calculated according to the following equation:Scavenging effect (%) = (1 − absorbance sample/absorbance control) × 100.

### 2.7. Strain Survival under Conditions Simulating the Human Gastrointestinal Tract

The resistance of selected strains under low pH conditions, similar to the conditions in the human gastrointestinal tract, was tested as previously described [32]. Briefly, bacterial cells obtained after 18 h of culture were harvested by centrifugation at 2000× *g* for 5 min at 4°C, and washed once with PBS, pH 7.4, before being resuspended (10^8^ CFU/mL) in the following solutions. To test resistance to pepsin and pancreatin, bacterial cells were resuspended into the simulated gastric fluid (SGF) and simulated intestinal fluid (SIF). SGF was prepared by supplementing sterilized PBS (pH 2, 3, and 4; adjusted with 1 N HCl) with pepsin (Sigma-Aldrich, St. Louis, MO, USA) to a final concentration of 3 g/L. SIF was prepared by supplementing sterilized PBS (pH 7 and 8; adjusted with 1 N NaOH) with pancreatin (Sigma-Aldrich, USA) to a final concentration of 1 g/L. Cells in SGF and SIF were incubated at 37 °C for 0, 1, 2, 3, and 4 h. The resistance of selected strains in every condition was evaluated by determining the viable colony count on MRS agar after treatment.

### 2.8. Strains Survival under Simulated Gastrointestinal Conditions

Simulated gastric and enteric juices were prepared according to Gänzle et al. [33], with modifications. The suspension of bacteria cellswas added to SGF and incubated at 37 °C for 120 min under agitation at 150 rpm (Innova 4000, New Brunswick Scientific, Enfield, CT, USA). In the next step, the pH of the juice was increased to 5.6 and pancreatin (1 g/L) (Sigma Chemical Co, Inc., USA) and bile salts (10 g/L) (Ox Gall Powder, Sigma Chemical Co, St. Louis, MO, USA) were added. The juice was incubated again at 37 °C for additional 120 min under agitation. In the last step, the pH was elevated to 7.5, the concentrations of pancreatin and bile salts were adjusted to 1 and 10 g/L, respectively, and juice was incubated again at 37 °C for an additional 120 min under agitation.

### 2.9. Antibiotic Sensitivity Test

Sensitivity to various antibiotics was measured using the disk diffusion method [34]. Bacterial suspensions were spread onto Muller-Hinton agar (Difco) plates onto which antibiotic disks (Oxoid, London, UK) were then placed. The plates were incubated at 37 °C for 18–24 h under anaerobic conditions. The diameter of the zone of inhibition around each disk was measured and recorded. Each bacterial isolate was classified as resistant (R), intermediately resistant (I), or sensitive (S), according to the guidelines of the Clinical and Laboratory Standards Institute [34]. The following antibiotics, with their concentrations given in parentheses, were tested: Ampicillin (AM; 10 μg), cefotaxime (CTX; 30 μg), cefotetan (CTT; 30 μg), cephalothin (CF; 30 μg), ciprofloxacin (CIP; 5 μg), cefepime (CEP; 30 μg), erythromycin (E; 15 μg), gentamicin (GM; 10 μg), kanamycin (K; 30 μg), nalidixic acid (NA; 30 μg), rifampicin (RA; 5 μg), tetracycline (TE; 30 μg), trimethoprim/sulfamethoxazole (SXT; 1.25 μg and 23.75 μg), and vancomycin (VA; 30 μg).

### 2.10. Statistical Analysis

Results are expressed as means ± SD of three experiments. Difference between groups was evaluated using the Student’s *t*-test, and a *p* value of <0.05 was considered to indicate a significant difference.

## 3. Results and Discussion

### 3.1. Inhibition of Isolated Strains on LPS-Induced NO Production

Isolated strains were screened for anti-inflammatory activity. NO has known as an inflammatory mediator in diverse infectious diseases. The physiological or normal NO production in phagocytes is beneficial for the host defense against microorganisms, parasites, and tumor cells [35]. To evaluate the effect of isolated strains on NO production in LPS-RAW 264.7 macrophages, nitrite accumulation was tested using the Griess assay. The negative control (*w*/*o* LPS) did not show inhibitory activity (5.5 ± 1.0 µM). Among the isolates, MG4261 showed the lowest nitrate production (11.5 ± 0.2 μM), followed by MG4231 (11.9 ± 0.6 μM), MG4270 (12.1 ± 0.4 μM), and MG4221 (12.3 ± 0.3 μM) (Figure 1).

### 3.2. Identification of Selected Strains

The four selected strains from the initial screening were identified based on 16S ribosomal RNA gene sequences. Based on a BLAST analysis, MG4221 and MG4270 showed sequence similarity to *Lactobacillus plantarum*. MG4231 and MG4261 showed similarity to *Lactobacillus fermentum* (Figure 2). Morita et al. [36] identified 10 *L. fermentum* strains that form nitrosomyoglobin in the absence of nitrite. ^15^N originating from l-[guanidino-^15^N_2_] arginine was converted to ^15^NO by the bacterial cells. Additionally, Hugas and Monfort [37] reported that *L. plantarum* strains reduce nitrate to nitrite via nitrate reductase in fermented meat products. The selected strains appeared as regular rods of about 1–1.5 µm long, with some shorter forms and an occasional elongated cell (Figure 3). Bacterial cells have a stress-like phenotype, probably due to the harvesting during the stationary phase [38,39].

### 3.3. DPPH Scavenging Activity

Antioxidant substances donate electrons or hydrogen atoms to free radicals to create a complex. Therefore, the antioxidant activity can be measured based on the electron-donating ability. The DPPH radical scavenging activities of culture filtrates of the selected strains were near and or greater than 50% (Figure 4). MG4270 showed the highest DPPH radical scavenging rates (65.81% ± 9.68%). The previous study reported that a non-purified solution of the culture supernatant that had over 50% antioxidant activity was demonstrating potential as a natural antioxidant [40]. Additionally, the antioxidant capacity could be increased by optimizing the environmental factors, which makes it possible to obtain useful industrial materials.

### 3.4. Resistance to Simulated Gastrointestinal Juices

To be able to exert beneficial effects, probiotic candidates should be able to maintain viability along the gastrointestinal transit by resisting to the harsh conditions [41]. The low pH of the stomach and the antimicrobial action of pepsin provide an effective barrier for the survival of bacteria into the gastrointestinal tract. The effects of SIF on the survival of the selected strains over different incubation periods are shown in Table 1. The population densities of all strains were <3 log CFU/mL after exposure to pH 2 for 3 h. Tolerance to bile salt and pancreatin is required for the survival of LAB in the small intestine [42]. The viable counts of selected strains were >8 log CFU/mL after exposure to SGF for 4 h, with low reductions in viable counts (<1 log CFU/mL). These results indicate that the selected strains are likely able to survive in the stomach and intestinal juices.

### 3.5. Antibiotic Sensitivity Test

The antibiotic resistance profiles of probiotic microorganisms are thought to be advantageous for survival in the gastrointestinal tract during medical treatment with an antibiotic [43,44]. However, Antibiotic resistance is present in different species of probiotic strains in commercially available food and drugs, which confirm the threat of spreading antibiotic resistance genes through the use of probiotics [45]. The sensitivity and resistance patterns that we obtained for the selected strains using 14 antibiotics are shown in Table 2. MG4261 was resistant to six antibiotics. All selected strains were resistant to vancomycin and streptomycin. Several species of *Lactobacillus* are intrinsically resistant to vancomycin [46]. The vancomycin resistance in some species of *Lactobacillus* spp. (e.g., *L. rhamnosus*, *L. casei*, *L. plantarum*, *L. fermentum*, *L. brevis*, and *L. curvatus*) is chromosomally encoded and not inducible or transferable [47]. Other studies reported that some *Lactobacillus* strains are naturally resistant to vancomycin and streptomycin, and such resistance is usually intrinsic, chromosomally encoded, and not transmissible, and are generally not associated with safety issues [47,48].

In conclusion, this study provides original information about isolated lactic acid bacteria that hadan interesting probiotic potential and, at the same time, exerted an antioxidant activity. The specific functions of the candidate probiotic bacteria identified in this study should be verified by evaluating physiological traits in animal models.

## Figures and Tables

**Figure 1 microorganisms-07-00109-f001:**
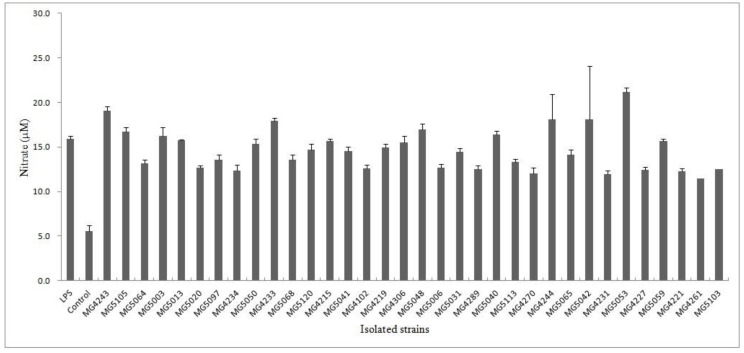
Inhibitory activity of isolated strains on NO production in LPS-induced RAW264.7 macrophages. NG-Monomethyl-L-arginine acetate (L-NMMA) was used as a positive control at a concentration of 10 μM. The results are presented as means ± SD of three independent experiments (*n* = 3).

**Figure 2 microorganisms-07-00109-f002:**
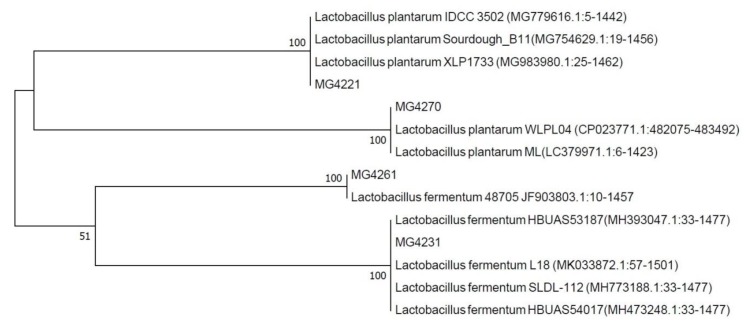
Neighbor-joining tree based on bacterial 16S rRNA gene sequence data, from different isolates obtained in the current study as well as sequences available in the GenBank database. Numerical values indicate bootstrap percentages from 1000 replicates.

**Figure 3 microorganisms-07-00109-f003:**
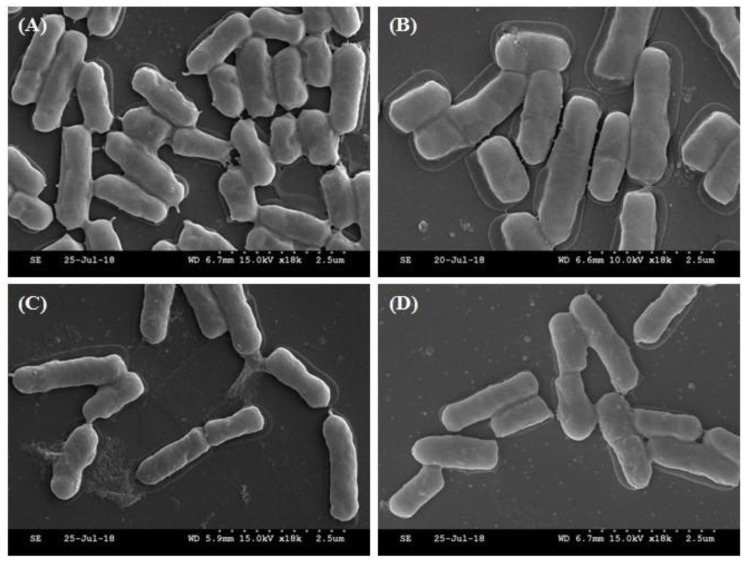
The scanning electron microscope (SEM)images of selected strains:(**A**) *Lactobacillus plantarum* MG4221; (**B**) *L. fermentum* MG4231; (**C**) *L. fermentum* MG4261; and(**D**) *L. plantarum* MG4270.

**Figure 4 microorganisms-07-00109-f004:**
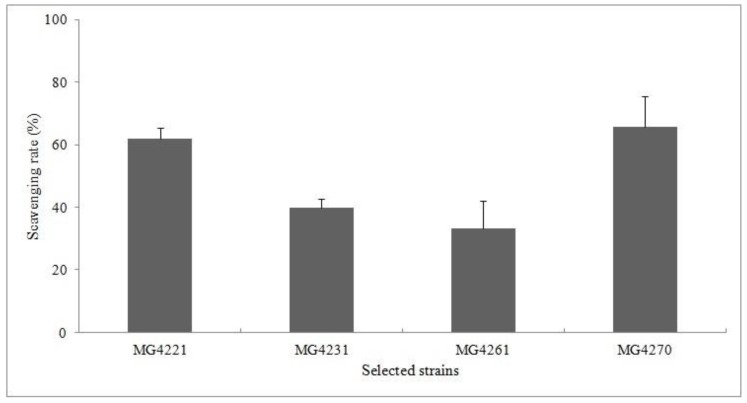
Scavenging rates of 2,2-Diphenyl-1-picrylhydrazyl (DPPH) radicals for selected strains.

**Table 1 microorganisms-07-00109-t001:** Survival of the selected *Lactobacillus* strains in simulated human gastrointestinal tract conditions.

Selected Strains	Simulated Gastric Fluid ^(1)^			Simulated Intestinal Fluid ^(2)^	
	pH2	pH3	pH4	pH7	pH8
MG4221	4.20 ± 0.03	6.86 ± 0.06	7.91 ± 0.06	9.01 ± 0.08	9.02 ± 0.04
MG4231	3.17 ± 0.02	6.02 ± 0.11	6.92 ± 0.07	9.04 ± 0.04	8.90 ± 0.01
MG4261	3.46 ± 0.07	6.73 ± 0.07	6.91 ± 0.08	7.98 ± 0.02	7.99 ± 0.05
MG4270	3.00 ± 0.02	5.92 ± 0.03	7.65 ± 0.04	9.00 ± 0.11	8.98 ± 0.06

^(1)^ Simulated gastric tolerance results are shown as viable counts (log CFU/mL) for each strain at pH 2, pH 3, and pH 4 after 3 h; ^(2)^ Simulated intestinal tolerance results are shown as viable counts (log CFU/mL) for each strain at 37 °C after 4 h.

**Table 2 microorganisms-07-00109-t002:** Antibiotic sensitivity and resistance of selected strains.

Antibiotics (µg/disc)	Selected Strains			
	MG4221	MG4231	MG4261	MG4270
Sulphamethoxazole-trimethoprim (1.25/23.75)	R	S	S	R
Tetracyclin (30)	I	S	S	S
Cephalothin (30)	S	S	S	S
Gentamicin (10)	S	I	R	S
Erythromycin (15)	S	S	S	S
Vancomycin (30)	R	R	R	R
Ampicillin (10)	S	S	S	S
Rifampicin (5)	S	S	S	S
Ciprofloxacin (5)	R	I	R	I
Cefotaxime (30)	S	S	S	S
Cefepime (30)	S	R	S	S
Cefotetan (30)	R	R	R	R
Nalidixic acid (30)	R	R	R	R
Kanamycin (30)	I	I	I	S

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
