# Peer review of "Inhibition of Nitric Oxide Production, Oxidative Stress Prevention, and Probiotic Activity of Lactic Acid Bacteria Isolated from the Human Vagina and Fermented Food"

_microorganisms, 2019, doi:10.3390/microorganisms7040109_

Reviewer 1 Report

The manuscript reporta a study on the probiotics properties on some strains isolated from women vagina. The data are interesting but discussion of the data needs to be implemented with other  references and comments, perhaps comparing the results with that obtained  on other strains or bacterial species (see paragraph 3.1, 3.3, 3.5 that are totally lacking in discussion of the data)

Other remark:

P4 L131: What is the normal or physiologial production of NO? Is it possible to insert here a reference number to be clearer?

Author Response

Microorganisms

Manuscript ID:      microorganisms-440400 - Revision

Manuscript      Title: Inhibition of nitric oxide      production and probiotic activity of lactic acid bacteria isolated from      human vagina and fermented food

Dear reviewers and editor,

We, authors would like to thank you for your effort and suggestions in order to improve our paper. We did our best for the corrections of the manuscript according to reviewers’ comments as a report attached below.

Major revisions:
1.      The English language quality is not adequate and in all the text are present several grammatical and construction errors, thus the MS requires for publication a deep revision from a native speaker, e.g. the sentence “Functional food products containing probiotics, also known as have several therapeutic benefits including anticancer, hypoglycemic, antioxidant, and immunomodulatory effects”.

Reply) Thank you for your comments.

The English language was checked in the manuscript

The sentence “Functional food products containing probiotics, also known as have several therapeutic benefits including anticancer, hypoglycemic, antioxidant, and immunomodulatory effects” was changed toFunctional foods containing probiotics, also known to have several therapeutic benefits including anticancer, hypoglycemic, antioxidant, and immunomodulatory effects

2.      The Authors should provide a rational for choosing fermented food and vaginas as source to isolate potential probiotic among LAB (more precisely lactobacilli). Moreover, nature of fermented food are not specified, neither age and other characteristics of the healthy donors (inclusion-exclusion criteria?). Where the swabs have been collected? Do the volunteers signed a consensus? If these isolations refer to a previous study, please state it more clearly.

Reply) Thank you for your comments.

The rational for choosing fermented food and vaginas were added in introduction part

Lactobacilli play an important role by producing compounds that inhibit the growth of potential pathogens such as lactic acid, bacteriocins and hydrogen peroxide [11]. The total bacteria isolated from healthy women showed at least 70% vaginal lactobacilli [12]. In the human vagina, the lactobacillus species have received considerable attention because of their protective and probiotic properties [13]. Moreover, the traditional fermented foods have been consumed by millions of people for thousands of years because of their beneficial properties. Most of these fermented foods are enriched with bacteria which has been well reported in previous studies [14, 15, 16]

3.      The Introduction is way too short and poorly informative.

Reply) Thank you for your comments.

More information was added in “Introduction” part

4.      Results regarding the resistance to the gastro-intestinal transit are not that useful if the simulated gastric juice and the simulated intestinal juice are not tested in sequence. Testing separately the resistance to low pH and the resistance to bile is not informative, neither predictive of the viability potential of bacteria along all the GIT.

Reply) Thank you for your comments.

More information of the gastrointestinal test was added

2.8 Strains Survival under Simulated Gastrointestinal Conditions

Simulated gastric and enteric juices were prepared according to Sallans et al. (1988) and Gänzle et al. (1999), with modifications. The suspension of bacteria cells was added to SGF and incubated at 37℃ for 120 min under agitation at 150 rpm (Innova 4000, New Brunswick Scientific, USA). In the next step, the pH of the juice was increased to 5.6 and pancreatin (1 g/L) (Sigma Chemical Co, Inc., USA) and bile salts (10 g/L) (Ox Gall Powder, Sigma Chemical Co) were added. The juice was incubated again at 37°C for additional 120 min under agitation. In the last step, the pH was elevated to 7.5, the concentrations of pancreatin and bile salts were adjusted to 1 g/L and 10 g/L, respectively, and juice was incubated again at 37℃ for an additional 120 min under agitation.  

5.      For the antibiotic susceptibility MIC test should be used instead of disc diffusion method.

Reply) Thank you for your comments.

Previous studies showed that the disc diffusion method is simple and practical and has been well standardized:

-            Jorgensen, J.H.; Turnidge, J.D. Antibacterial susceptibility tests: dilution and disk diffusion methods. In: Murray PR, Baron EJ, Jorgensen JH, Landry ML, Pfaller MA, eds. Manual of clinical microbiology. 9th ed. Washington, DC: American Society for Microbiology. 2007, 1152–1172.

-            Bauer, A.W.; Kirby, W.M.M.; Sherris, J.C.; Turk, M. Antibiotic susceptibility testing by a standardized single disk method. Am J Clin Pathol. 1966, 45,493–496.

-            Clinical and Laboratory Standards Institute. Performance standards for antimicrobial disk susceptibility tests. Approved standard M2–A10. Wayne, PA: Clinical and Laboratory Standards Institute, 2009.

6.      Lane 134: it is not clear what the Authors mean with “The negative control (w/o LPS) did not show inhibitory activity”. The control condition is meant to show the basal level of NO production in absence of a proinflammatory stimulus like LPS, therefore it is not clear the comment on the lack of inhibitory activity.

Reply) Thank you for your comments.

7.      DPPH scavenging activity: specify which condition has been used to calculate the % activity showed in Fig. 4 legend. And please clarify the sentence: “…the culture supernatant had 50% antioxidant activity as a non-purified solution, demonstrating potential as a natural antioxidant.”

Reply) Thank you for your comments.

The condition has been used to calculate the % activity showed in Fig. 4 legend was showed in 2.6. of “Materials and Methods” part:

After incubation at room temperature for 30 min, the absorbance was measured at 517 nm. Two milliliters of methanol mixed with 2 ml of DPPH solution was used as the control. The inhibition of DPPH (expressed as a percentage) was calculated according to the following equation:

Scavenging effect (%) = (1 – absorbance sample/absorbance control) × 100.

The sentence: “...the culture supernatant had 50% antioxidant activity as a non-purified solution, demonstrating potential as a natural antioxidant.” was changed to “The previous study reported that a non-purified solution of the culture supernatant had over 50% antioxidant activity was demonstrating potential as a natural antioxidant”

8.      The Authors should provide a more detailed explanation about the sentence: “The antibiotic resistance profiles of probiotic microorganisms are thought to be advantageous for survival in the gastrointestinal tract during medical treatment with an antibiotic”, since the presence of antibiotic resistance in bacteria is something undesirable, that also represents a concern (https://www.ncbi.nlm.nih.gov/pmc/articles/PMC5156686/). Please consider and discuss this aspect, that should be independent from the comment on intrinsic resistances.

Reply) Thank you for your comments.

A more detailed explanations were added

“The antibiotic resistance profiles of probiotic microorganisms are thought to be advantageous for survival in the gastrointestinal tract during medical treatment with an antibiotic [35, 36]. However, Antibiotic resistance is present in different species of probiotic strains in commercially available food and drugs which confirm the threat of spreading antibiotic resistance genes through the use of probiotics [37]”

9.      The sentence: “In conclusion, this study provides a basis for the formulation of novel probiotic foods or supplements that promote the prevention of oxidative stress and related diseases.” is way too rash, as it is based only on few in vitro characterizations, not properly performed. I invite Authors to be more cautious and mild in such statement.

Reply) Thank you for your comments.

The sentence was changed to In conclusion, this study provides basic information about isolated lactic acid bacteria which have high probiotic potentiality in exerting antioxidant activity”

Minor revisions:

1.      Lane 25: please report, in the text and in the References section, also the official definition of probiotics by EFSA/WHO.

Reply) Thank you for your comments.

The official definition of probiotics by EFSA/WHO was reported in the manuscript

2.      Lane 27: please correct “diarrheal” with “diarrhea”

Reply) Thank you for your comments.

The word “diarrheal” was changed to “diarrhea”

3.      Lanes 36-37: Authors claim: “In chronically inflamed tissues, the stimulus is persistent”; however, this is not always true, as chronic inflammation can be caused, even not anymore in presence of the trigger, by an over and/or prolonged reaction.  Please provide at least one reference for such sentences.

Reply) Thank you for your comments.

The sentence In chronically inflamed tissues, the stimulus is persistent” was changed to “Persistent inflammatory stimuli can lead to chronic inflammation” and the reference was also added:

Vincenzo, B.; Asif, I.J.; Nikolaos, P.; Francesco, M. Adaptive Immunity and Inflammation. Int J Inflam 2015, Article ID 575406, 1 page. Available online: http://dx.doi.org/10.1155/2015/575406

4.      Lanes 44-45: I would change the sentence: “The purpose of this study was to isolate lactic acid bacteria (LAB) with antioxidant activity and accordingly high probiotic potential” as it is not clear what the Authors mean with “accordingly”, since the antioxidant potential is not the sole condition for a strain to be claimed as probiotic.

Reply) Thank you for your comments.

The sentence was changed to “The aim of this study was to isolate lactic acid bacteria with high probiotic potentiality in exerting antioxidant activity for counteracting oxidative stress

5.      Lane 50: please correct “health Korean women” with “healthy Korean women”

Reply) Thank you for your comments.

The phrase “health Korean women” was changed to “healthy Korean women

6.      Please change all over the manuscript “mL” with “ml”.

Reply) Thank you for your comments.

The word “mL” was changed to “ml” all over the manuscript.

7.      Lane 52-53: correct “Man Rogosa Sharpe” with “de Man Rogosa Sharpe”. This medium is however selective, within LAB, for lactobacilli.

Reply) Thank you for your comments.

Man Rogosa Sharpe” was changed to “de Man Rogosa Sharpe”

8.      Why the Authors did not add L-glutamine to DMEM medium used to cultivate macrophages?

Reply) Thank you for your comments.

According to previous study, macrophages cells were maintained and cultured in DMEM medium without adding L-glutamine. The reference was attached

Kawakami, T.; Kawamura, K.; Fujimori, K.; Koike, A.; Amano F. Influence of the culture medium on the production of nitric oxide and expression of inducible nitric oxide synthase by activated macrophages in vitro. Biochem Biophys Rep. 2016, 5, 328-334

9.      Lane 101: is it possible to pellet bacterial cells at “2000 × g”, or is it a type error?

Reply) Thank you for your comments.

The bacteria were desired to be used for the next experiment. Therefore, if harvesting at high rpm would damage the growing cell wall and membrane due to higher shear and stress exerted by higher centrifuge force. Therefore, the centrifuge with 2000 × g was applied for harvesting bacteria cells in this experiment.

10.  Lane 129-130: this phrase is not clear, please revise it: “NO is a multi-functional mediator and an important part of the immune response to inflammatory activity.”

Reply) Thank you for your comments.

The sentence was changed to “NO has known as an inflammatory mediator in diverse infectious diseases

11.  Lane 133: “LPS-induced RAW 264.7 macrophages”, use “LPS- RAW 264.7 macrophages” instead.

Reply) Thank you for your comments.

“LPS- RAW 264.7 macrophages” was used instead of “LPS-induced RAW 264.7 macrophages”

12.  Figure 1 legend: please specify what “L-NMMA” stands for, and also specify what is the Control condition, namely only media (without LPS).

Reply) Thank you for your comments.

13.  Lane 165-166: Please re-formulate the sentence: “The increase in antioxidant capacity by the optimization of environmental factors may make it possible to obtain useful industrial materials”.

Reply) Thank you for your comments.

The sentence was re-formulated

Additionally, the antioxidant capacity could be increased by optimizing the environmental factors which makes it possible to obtain useful industrial materials.

14.  Lane 170-171: Instead of: “For health benefits, lactobacilli need to resist the harsh conditions of the stomach and upper intestine” I would rather say: “To be able to exert beneficial effects, probiotic candidates should be able to maintain viability along the gastrointestinal transit by resisting to the harsh conditions…..”

Reply) Thank you for your comments.

The sentence was changed

15.  Lane 172: substitute “entry” with “survival”.

Reply) Thank you for your comments.

The word “survival” was changed to “entry”

16.  Lane 175: Add “for” before “the survival”.

Reply) Thank you for your comments.

The word “for” was added

17.  Lane 177: add “able” after “likely”.

Reply) Thank you for your comments.

The word “able” was added

18.  Table 1 legend: I would substitute “Survival of lactic acid bacteria in simulated human gastrointestinal tract conditions” with “Survival of the selected Lactobacillus strains in…”

Reply) Thank you for your comments.

The sentence was changed in Table 1 legend

Reviewer 2 Report

This manuscript describe the inhibitory activity of LAB strains against nitric oxide production. The authors should integrate the section results and discussion, improving information on scavenging activity. 

Author Response

(The authors gave the same response as above.)

Reviewer 3 Report

General remarks

The manuscript by Kang et al. described isolation and characterization of lactobacilli isolated from different sources and evaluated for inhibition of NO production on a macrophage cell line, scavenging capacity, antibiotic ang gastro-intestinal stresses resistance. The manuscript as it is in this form cannot be accepted for publications, unless all the following observation will be addressed:

Major revisions

1.      The English language quality is not adequate and in all the text are present several grammatical and construction errors, thus the MS requires for publication a deep revision from a native speaker, e.g. the sentence “Functional food products containing probiotics, also known as have several therapeutic benefits including anticancer, hypoglycemic, antioxidant, and immunomodulatory effects”.

2.      The Authors should provide a rational for choosing fermented food and vaginas as source to isolate potential probiotic among LAB (more precisely lactobacilli). Moreover, nature of fermented food are not specified, neither age and other characteristics of the healthy donors (inclusion-exclusion criteria?). Where the swabs have been collected? Do the volunteers signed a consensus? If these isolations refer to a previous study, please state it more clearly.

3.      The Introduction is way too short and poorly informative.

4.      Results regarding the resistance to the gastro-intestinal transit are not that useful if the simulated gastric juice and the simulated intestinal juice are not tested in sequence. Testing separately the resistance to low pH and the resistance to bile is not informative, neither predictive of the viability potential of bacteria along all the GIT.

5.      For the antibiotic susceptibility MIC test should be used instead of disc diffusion method.

6.      Lane 134: it is not clear what the Authors mean with “The negative control (w/o LPS) did not show inhibitory activity”. The control condition is meant to show the basal level of NO production in absence of a proinflammatory stimulus like LPS, therefore it is not clear the comment on the lack of inhibitory activity.

7.      DPPH scavenging activity: specify which condition has been used to calculate the % activity showed in Fig. 4 legend. And please clarify the sentence: “..the culture supernatant had 50% antioxidant activity as a non-purified solution, demonstrating potential as a natural antioxidant.”

8.      The Authors should provide a more detailed explanation about the sentence: “The antibiotic resistance profiles of probiotic microorganisms are thought to be advantageous for survival in the gastrointestinal tract during medical treatment with an antibiotic”, since the presence of antibiotic resistance in bacteria is something undesirable, that also represents a concern (https://www.ncbi.nlm.nih.gov/pmc/articles/PMC5156686/). Please consider and discuss this aspect, that should be independent from the comment on intrinsic resistances.

9.      The sentence: “In conclusion, this study provides a basis for the formulation of novel probiotic foods or supplements that promote the prevention of oxidative stress and related diseases.” is way too rash, as it is based only on few in vitro characterizations, not properly performed. I invite Authors to be more cautious and mild in such statement.

Minor revisions

1.      Lane 25: please report, in the text and in the References section, also the official definition of probiotics by EFSA/WHO.

2.      Lane 27: please correct “diarrheal” with “diarrhea”

3.      Lanes 36-37: Authors claim: “In chronically inflamed tissues, the stimulus is persistent”; however, this is not always true, as chronic inflammation can be caused, even not anymore in presence of the trigger, by an over and/or prolonged reaction.  Please provide at least one reference for such sentences.

4.      Lanes 44-45: I would change the sentence: “The purpose of this study was to isolate lactic acid bacteria (LAB) with antioxidant activity and accordingly high probiotic potential” as it is not clear what the Authors mean with “accordingly”, since the antioxidant potential is not the sole condition for a strain to be claimed as probiotic.

5.      Lane 50: please correct “health Korean women” with “healthy Korean women”

6.      Please change all over the manuscript “mL” with “ml”.

7.      Lane 52-53: correct “Man Rogosa Sharpe” with “de Man Rogosa Sharpe”. This medium is however selective, within LAB, for lactobacilli.

8.      Why the Authors did not add L-glutamine to DMEM medium used to cultivate macrophages?

9.      Lane 101: is it possible to pellet bacterial cells at “2000 × g”, or is it a type error?

10.  Lane 129-130: this phrase is not clear, please revise it: “NO is a multi-functional mediator and an important part of the immune response to inflammatory activity.”

11.  Lane 133: “LPS-induced RAW 264.7 macrophages”, use “LPS- RAW 264.7 macrophages” instead.

12.  Figure 1 legend: please specify what “L-NMMA” stands for, and also specify what is the Control condition, namely only media (without LPS).

13.  Lane 165-166: Please re-formulate the sentence: “The increase in antioxidant capacity by the optimization of environmental factors may make it possible to obtain useful industrial materials”.

14.  Lane 170-171: Instead of: “For health benefits, lactobacilli need to resist the harsh conditions of the stomach and upper intestine” I would rather say: “To be able to exert beneficial effects, probiotic candidates should be able to maintain viability along the gastrointestinal transit by resisting to the harsh conditions…..”

15.  Lane 172: substitute “entry” with “survival”.

16.  Lane 175: Add “for” before “the survival”.

17.  Lane 177: add “able” after “likely”.

18.  Table 1 legend: I would substitute “Survival of lactic acid bacteria in simulated human gastrointestinal tract conditions” with “Survival of the selected Lactobacillus strains in…”

Author Response

Microorganisms

Manuscript ID:      microorganisms-440400 - Revision

Manuscript      Title: Inhibition of nitric oxide      production and probiotic activity of lactic acid bacteria isolated from      human vagina and fermented food

Dear reviewers and editor,

We, authors would like to thank you for your effort and suggestions in order to improve our paper. We did our best for the corrections of the manuscript according to reviewers’ comments as a report attached below.

Major revisions:
1.      The English language quality is not adequate and in all the text are present several grammatical and construction errors, thus the MS requires for publication a deep revision from a native speaker, e.g. the sentence “Functional food products containing probiotics, also known as have several therapeutic benefits including anticancer, hypoglycemic, antioxidant, and immunomodulatory effects”.

Reply) Thank you for your comments.

The English language was checked in the manuscript

The sentence “Functional food products containing probiotics, also known as have several therapeutic benefits including anticancer, hypoglycemic, antioxidant, and immunomodulatory effects” was changed toFunctional foods containing probiotics, also known to have several therapeutic benefits including anticancer, hypoglycemic, antioxidant, and immunomodulatory effects

2.      The Authors should provide a rational for choosing fermented food and vaginas as source to isolate potential probiotic among LAB (more precisely lactobacilli). Moreover, nature of fermented food are not specified, neither age and other characteristics of the healthy donors (inclusion-exclusion criteria?). Where the swabs have been collected? Do the volunteers signed a consensus? If these isolations refer to a previous study, please state it more clearly.

Reply) Thank you for your comments.

The rational for choosing fermented food and vaginas were added in introduction part

Lactobacilli play an important role by producing compounds that inhibit the growth of potential pathogens such as lactic acid, bacteriocins and hydrogen peroxide [11]. The total bacteria isolated from healthy women showed at least 70% vaginal lactobacilli [12]. In the human vagina, the lactobacillus species have received considerable attention because of their protective and probiotic properties [13]. Moreover, the traditional fermented foods have been consumed by millions of people for thousands of years because of their beneficial properties. Most of these fermented foods are enriched with bacteria which has been well reported in previous studies [14, 15, 16]

3.      The Introduction is way too short and poorly informative.

Reply) Thank you for your comments.

More information was added in “Introduction” part

4.      Results regarding the resistance to the gastro-intestinal transit are not that useful if the simulated gastric juice and the simulated intestinal juice are not tested in sequence. Testing separately the resistance to low pH and the resistance to bile is not informative, neither predictive of the viability potential of bacteria along all the GIT.

Reply) Thank you for your comments.

More information of the gastrointestinal test was added

2.8 Strains Survival under Simulated Gastrointestinal Conditions

Simulated gastric and enteric juices were prepared according to Sallans et al. (1988) and Gänzle et al. (1999), with modifications. The suspension of bacteria cells was added to SGF and incubated at 37℃ for 120 min under agitation at 150 rpm (Innova 4000, New Brunswick Scientific, USA). In the next step, the pH of the juice was increased to 5.6 and pancreatin (1 g/L) (Sigma Chemical Co, Inc., USA) and bile salts (10 g/L) (Ox Gall Powder, Sigma Chemical Co) were added. The juice was incubated again at 37°C for additional 120 min under agitation. In the last step, the pH was elevated to 7.5, the concentrations of pancreatin and bile salts were adjusted to 1 g/L and 10 g/L, respectively, and juice was incubated again at 37℃ for an additional 120 min under agitation.  

5.      For the antibiotic susceptibility MIC test should be used instead of disc diffusion method.

Reply) Thank you for your comments.

Previous studies showed that the disc diffusion method is simple and practical and has been well standardized:

-            Jorgensen, J.H.; Turnidge, J.D. Antibacterial susceptibility tests: dilution and disk diffusion methods. In: Murray PR, Baron EJ, Jorgensen JH, Landry ML, Pfaller MA, eds. Manual of clinical microbiology. 9th ed. Washington, DC: American Society for Microbiology. 2007, 1152–1172.

-            Bauer, A.W.; Kirby, W.M.M.; Sherris, J.C.; Turk, M. Antibiotic susceptibility testing by a standardized single disk method. Am J Clin Pathol. 1966, 45,493–496.

-            Clinical and Laboratory Standards Institute. Performance standards for antimicrobial disk susceptibility tests. Approved standard M2–A10. Wayne, PA: Clinical and Laboratory Standards Institute, 2009.

6.      Lane 134: it is not clear what the Authors mean with “The negative control (w/o LPS) did not show inhibitory activity”. The control condition is meant to show the basal level of NO production in absence of a proinflammatory stimulus like LPS, therefore it is not clear the comment on the lack of inhibitory activity.

Reply) Thank you for your comments.

7.      DPPH scavenging activity: specify which condition has been used to calculate the % activity showed in Fig. 4 legend. And please clarify the sentence: “…the culture supernatant had 50% antioxidant activity as a non-purified solution, demonstrating potential as a natural antioxidant.”

Reply) Thank you for your comments.

The condition has been used to calculate the % activity showed in Fig. 4 legend was showed in 2.6. of “Materials and Methods” part:

After incubation at room temperature for 30 min, the absorbance was measured at 517 nm. Two milliliters of methanol mixed with 2 ml of DPPH solution was used as the control. The inhibition of DPPH (expressed as a percentage) was calculated according to the following equation:

Scavenging effect (%) = (1 – absorbance sample/absorbance control) × 100.

The sentence: “...the culture supernatant had 50% antioxidant activity as a non-purified solution, demonstrating potential as a natural antioxidant.” was changed to “The previous study reported that a non-purified solution of the culture supernatant had over 50% antioxidant activity was demonstrating potential as a natural antioxidant”

8.      The Authors should provide a more detailed explanation about the sentence: “The antibiotic resistance profiles of probiotic microorganisms are thought to be advantageous for survival in the gastrointestinal tract during medical treatment with an antibiotic”, since the presence of antibiotic resistance in bacteria is something undesirable, that also represents a concern (https://www.ncbi.nlm.nih.gov/pmc/articles/PMC5156686/). Please consider and discuss this aspect, that should be independent from the comment on intrinsic resistances.

Reply) Thank you for your comments.

A more detailed explanations were added

“The antibiotic resistance profiles of probiotic microorganisms are thought to be advantageous for survival in the gastrointestinal tract during medical treatment with an antibiotic [35, 36]. However, Antibiotic resistance is present in different species of probiotic strains in commercially available food and drugs which confirm the threat of spreading antibiotic resistance genes through the use of probiotics [37]”

9.      The sentence: “In conclusion, this study provides a basis for the formulation of novel probiotic foods or supplements that promote the prevention of oxidative stress and related diseases.” is way too rash, as it is based only on few in vitro characterizations, not properly performed. I invite Authors to be more cautious and mild in such statement.

Reply) Thank you for your comments.

The sentence was changed to In conclusion, this study provides basic information about isolated lactic acid bacteria which have high probiotic potentiality in exerting antioxidant activity”

Minor revisions:

1.      Lane 25: please report, in the text and in the References section, also the official definition of probiotics by EFSA/WHO.

Reply) Thank you for your comments.

The official definition of probiotics by EFSA/WHO was reported in the manuscript

2.      Lane 27: please correct “diarrheal” with “diarrhea”

Reply) Thank you for your comments.

The word “diarrheal” was changed to “diarrhea”

3.      Lanes 36-37: Authors claim: “In chronically inflamed tissues, the stimulus is persistent”; however, this is not always true, as chronic inflammation can be caused, even not anymore in presence of the trigger, by an over and/or prolonged reaction.  Please provide at least one reference for such sentences.

Reply) Thank you for your comments.

The sentence In chronically inflamed tissues, the stimulus is persistent” was changed to “Persistent inflammatory stimuli can lead to chronic inflammation” and the reference was also added:

Vincenzo, B.; Asif, I.J.; Nikolaos, P.; Francesco, M. Adaptive Immunity and Inflammation. Int J Inflam 2015, Article ID 575406, 1 page. Available online: http://dx.doi.org/10.1155/2015/575406

4.      Lanes 44-45: I would change the sentence: “The purpose of this study was to isolate lactic acid bacteria (LAB) with antioxidant activity and accordingly high probiotic potential” as it is not clear what the Authors mean with “accordingly”, since the antioxidant potential is not the sole condition for a strain to be claimed as probiotic.

Reply) Thank you for your comments.

The sentence was changed to “The aim of this study was to isolate lactic acid bacteria with high probiotic potentiality in exerting antioxidant activity for counteracting oxidative stress

5.      Lane 50: please correct “health Korean women” with “healthy Korean women”

Reply) Thank you for your comments.

The phrase “health Korean women” was changed to “healthy Korean women

6.      Please change all over the manuscript “mL” with “ml”.

Reply) Thank you for your comments.

The word “mL” was changed to “ml” all over the manuscript.

7.      Lane 52-53: correct “Man Rogosa Sharpe” with “de Man Rogosa Sharpe”. This medium is however selective, within LAB, for lactobacilli.

Reply) Thank you for your comments.

Man Rogosa Sharpe” was changed to “de Man Rogosa Sharpe”

8.      Why the Authors did not add L-glutamine to DMEM medium used to cultivate macrophages?

Reply) Thank you for your comments.

According to previous study, macrophages cells were maintained and cultured in DMEM medium without adding L-glutamine. The reference was attached

Kawakami, T.; Kawamura, K.; Fujimori, K.; Koike, A.; Amano F. Influence of the culture medium on the production of nitric oxide and expression of inducible nitric oxide synthase by activated macrophages in vitro. Biochem Biophys Rep. 2016, 5, 328-334

9.      Lane 101: is it possible to pellet bacterial cells at “2000 × g”, or is it a type error?

Reply) Thank you for your comments.

The bacteria were desired to be used for the next experiment. Therefore, if harvesting at high rpm would damage the growing cell wall and membrane due to higher shear and stress exerted by higher centrifuge force. Therefore, the centrifuge with 2000 × g was applied for harvesting bacteria cells in this experiment.

10.  Lane 129-130: this phrase is not clear, please revise it: “NO is a multi-functional mediator and an important part of the immune response to inflammatory activity.”

Reply) Thank you for your comments.

The sentence was changed to “NO has known as an inflammatory mediator in diverse infectious diseases

11.  Lane 133: “LPS-induced RAW 264.7 macrophages”, use “LPS- RAW 264.7 macrophages” instead.

Reply) Thank you for your comments.

“LPS- RAW 264.7 macrophages” was used instead of “LPS-induced RAW 264.7 macrophages”

12.  Figure 1 legend: please specify what “L-NMMA” stands for, and also specify what is the Control condition, namely only media (without LPS).

Reply) Thank you for your comments.

13.  Lane 165-166: Please re-formulate the sentence: “The increase in antioxidant capacity by the optimization of environmental factors may make it possible to obtain useful industrial materials”.

Reply) Thank you for your comments.

The sentence was re-formulated

Additionally, the antioxidant capacity could be increased by optimizing the environmental factors which makes it possible to obtain useful industrial materials.

14.  Lane 170-171: Instead of: “For health benefits, lactobacilli need to resist the harsh conditions of the stomach and upper intestine” I would rather say: “To be able to exert beneficial effects, probiotic candidates should be able to maintain viability along the gastrointestinal transit by resisting to the harsh conditions…..”

Reply) Thank you for your comments.

The sentence was changed

15.  Lane 172: substitute “entry” with “survival”.

Reply) Thank you for your comments.

The word “survival” was changed to “entry”

16.  Lane 175: Add “for” before “the survival”.

Reply) Thank you for your comments.

The word “for” was added

17.  Lane 177: add “able” after “likely”.

Reply) Thank you for your comments.

The word “able” was added

18.  Table 1 legend: I would substitute “Survival of lactic acid bacteria in simulated human gastrointestinal tract conditions” with “Survival of the selected Lactobacillus strains in…”

Reply) Thank you for your comments.

The sentence was changed in Table 1 legend

Round  2

Reviewer 1 Report

Authors followed all the reviewer's suggestions, so in my opinion the manuscript is ready to be published

Author Response

Microorganisms

Manuscript ID:      microorganisms-440400 – minor revision

Manuscript      Title : Inhibition of nitric oxide      production and probiotic activity of lactic acid bacteria isolated from      human vagina and fermented food

Dear reviewers and editor,

We, authors would like to thank you for your effort and suggestions in order to improve our paper. We did our best for the corrections of the manuscript according to reviewer’ comments as a report attached below.

Minor revisions:
Line 2: Inhibition of nitric oxide production, oxidative stress prevention and…

Reply) Thank you for your comments.

The sentence was changed in manuscript

Line 18: potentially applicable

Reply) Thank you for your comments.

The word “potentially” was added

Line 27: benefit on the host” [3]. Please cite also https://www.mdpi.com/2076-2607/7/3/83/htm

Reply) Thank you for your comments.

The reference was cited in the manuscript:

4. Fenster, Kurt.; Freeburg, B.; Hollard, C.; Wong, C.; Laursen, R.R.; Ouwehand, A.C. The production and delivery of probiotics: A review of a practical approach. Microorganisms 2019, 7(3), 83; DOI: 10.3390/microorganisms7030083

Lines 33-34: several therapeutic benefits have been addressed to probiotics-based functional foods, e.g.

Reply) Thank you for your comments.

The sentence was rephrased

Lines 34-35: please include diabetes, allergic deseases and obesity, citing https://www.mdpi.com/2076-2607/7/3/67, https://www.mdpi.com/2227-9067/6/2/24 , https://www.mdpi.com/2072-6643/11/3/635

Reply) Thank you for your comments.

The references were cited in the manuscript

12. Mishra, S.; Wang, S.; Nagpal, R., Miller, B.; Singh, R.; Taraphder, S.; Yadav, H. Probiotics and prebiotics for the amelioration of type 1 diabetes: Present and future perspectives. Microorganisms 2019, 7(3), 67; Available online: https://doi.org/10.3390/microorganisms7030067

13. Wang, H.T.; Anvari, S.; Anagnostou, K. The role of probiotics in preventing allergic disease. Children 2019, 6(2), 24; Available online: https://doi.org/10.3390/children6020024

14. Cerdó, T.; García-Santos, J.A.; Bermúdez, M.G.; Campoy, C. The role of probiotics and prebiotics in the prevention and treatment of obesity. Nutrients 2019, 11(3), 635; Available online: https://doi.org/10.3390/nu11030635

Line 38: pathogens such as lactic acid, bacteriocins and hydrogen peroxide [11]. Please cite also https://www.mdpi.com/1420-3049/22/8/1255, https://www.mdpi.com/2304-8158/6/12/110/htm

Reply) Thank you for your comments.

The references were cited in the manuscript

16. Mokoena, M.P. Lactic acid bacteria and their bacteriocins: Classification, biosynthesis and applications against uropathogens: A mini-review. Molecules 2017, 22(8), 1255; Available online: https://doi.org/10.3390/molecules22081255

17. Russo, P.; Fares, C.; Longo, A.; Spano, G.; Capozzi V. Lactobacillus plantarum with broad antifungal activity as a protective starter culture for bread production. Foods 2017, 6(12), 110; DOI:10.3390/foods6120110

Lines 42-43: Most of the spontaneous fermentations associated to these foods included a crucial role of Lactobacillus strains () cite also https://www.mdpi.com/2311-5637/3/4/49

Reply) Thank you for your comments.

The reference was cited in the manuscript

20. Capozzi, V.; Fragasso, M.; Romaniello, R.; Berbegal, C.; Russo, P.; Spano, G. Spontaneous food fermentations and potential risks for human health. Fermentation 2017, 3(4), 49; Available online: https://doi.org/10.3390/fermentation3040049

Line 165: Bacterial cells have a stress-like phenotype, probably due to the harvesting during the stationary phase (). Please cite https://www.sciencedirect.com/science/article/pii/S0923250811000349 , https://www.tandfonline.com/doi/abs/10.1080/10408398.2019.1580673

Reply) Thank you for your comments.

The reference was cited in the manuscript

38. Capozzi, V.; Weidmann, S.; Fiocco, D.; Reiu, A.; Hols, P.; Guzzo, J.; Spano, G. Inactivation of a small heat shock protein affects cell morphology and membrane fluidity in Lactobacillus plantarum WCFS1. Res. Microbiol. 2011, 162(4), 419-425

39. Fiocco, D.; Longo, A.; Arena, M.P.; Russo, P.; Spano, G.; Capozzi, V. How probiotics face food stress: They get by with a little help, Crit Rev Food Sci Nutr 2019 DOI: 10.1080/10408398.2019.1580673

- Lines 215-216: In conclusion, this study provides original information about isolated lactic acid bacteria which have an interesting probiotic potential and, at the same time, exerted an antioxidant activity. This specific functions of the candidate probiotic

Reply) Thank you for your comments.

The sentences were rephrased:

In conclusion, this study provides original information about isolated lactic acid bacteria which have an interesting probiotic potential and, at the same time, exerted an antioxidant activity. This specific functions of the candidate probiotic bacteria identified in this study should be verified by evaluating physiological traits in animal models

- As required by reviewer 3, please provide at least a general description of the following issue: nature of fermented food are not specified, neither age and other characteristics of the healthy donors (inclusion-exclusion criteria?). Where the swabs have been collected? Do the volunteers signed a consensus? If these isolations refer to a previous study, please state it more clearly.

Reply) Thank you for your comments.

According to previous study, the clinical samples were obtained from Ewha Woman’s University Mokdong Hospital in Seoul. All samples were taken from patients who consented to the sampling potentially applicable. The reference has been cited in the manuscript:

27. Chang, C.E.; Pavlova, S.I.; Tao, L.; Kim, E.; Kim, S.C.; Yun, H.S.; So, J.S. Molecular identification of vaginal Lactobacillus spp. isolated from Korean women. J. Microbiol. Biotechnol. 2002, 12, 312‒317.

Reviewer 2 Report

 I was able to see the progress of this work. Therefore, I'd like to agree with publication of this work.  

Author Response

(The authors gave the same response as above.)

Reviewer 3 Report

Authors' attempts to satisfyingly reply to Reviewer's observation are limited and not convincing. In this form MS cannot be accepted. In the attached file replies and explanations.

Author Response

(The authors gave the same response as above.)
